# Full integration of highly stretchable inorganic transistors and circuits within molecular-tailored elastic substrates on a large scale

Seung-Han Kang [1,2,9], Jeong-Wan Jo [3,9], Jong Min Lee [1,2], Sanghee Moon[2], Seung Bum Shin[4], Su Bin Choi[5], Donghwan Byeon[1,2], Jaehyun Kim[6], Myung-Gil Kim [4], Yong-Hoon Kim [4] ✉, Jong-Woong Kim [5,7,8] ✉ & Sung Kyu Park [1,2] ✉

The emergence of high-form-factor electronics has led to a demand for high-density integration of inorganic thin-film devices and circuits with full stretchability. However, the intrinsic stiffness and brittleness of inorganic materials have impeded their utilization in free-form electronics. Here, we demonstrate highly integrated strain-insensitive stretchable metal-oxide transistors and circuitry (442 transistors/cm²) via a photolithography-based bottom-up approach, where transistors with fluidic liquid metal interconnection are embedded in large-area molecular-tailored heterogeneous elastic substrates (5 × 5 cm²). Amorphous indium-gallium-zinc-oxide transistor arrays (7 × 7), various logic gates, and ring-oscillator circuits exhibited strain-resilient properties with performance variation less than 20% when stretched up to 50% and 30% strain (10,000 cycles) for unit transistor and circuits, respectively. The transistors operate with an average mobility of 12.7 ($\pm$1.7) cm² V⁻¹s⁻¹, on/off current ratio of >10⁷, and the inverter, NAND, NOR circuits operate quite logically. Moreover, a ring oscillator comprising 14 cross-wired transistors validated the cascading of the multiple stages and device uniformity, indicating an oscillation frequency of ~70 kHz.

The growing demand for high-form-factor electronics has previously vitalized commercialization of flexible electronic devices, leading to extensive research on the design of deformable systems[1–3]. Beyond the flexible electronics, stretchable devices have received much attention owing to their great potential in the next-generation free-form electronic applications including soft robotics[4,5], free-form displays[6,7], wearable electronics[8,9], machine-neural interfaces[10], wearable energy harvesting[11–14], and biomedical healthcare systems[15–18]. A key

¹Department of Intelligent Semiconductor Engineering, Chung-Ang University, Seoul 06974, Korea. ²School of Electrical and Electronic Engineering, Chung-Ang University, Seoul 06974, Korea. ³Electrical Engineering Division, Department of Engineering, University of Cambridge, 9 JJ Thomson Avenue, Cambridge CB3 0FA, UK. ⁴School of Advanced Materials Science and Engineering, Sungkyunkwan University, Suwon 16419, Korea. ⁵Department of Smart Fab. Technology, Sungkyunkwan University, Suwon 16419, Korea. ⁶Department of Semiconductor Science, Dongguk University, Seoul 04620, Republic of Korea. ⁷School of Mechanical Engineering, Sungkyunkwan University, Suwon 16419, Korea. ⁸Department of Semiconductor Convergence Engineering, Sungkyunkwan University, Suwon 16419, Korea. ⁹These authors contributed equally: Seung-Han Kang, Jeong-Wan Jo. ✉e-mail: yhkim76@skku.edu; wyjd@skku.edu; skpark@cau.ac.kr

requirement for the realization of practical stretchable electronics capable of performing electrical functions under severe mechanical deformations and having great conformability to an arbitrary complex-shaped surface is to secure high-form-factor electronic devices with excellent electrical performance and stability[19]. For this reason, the application of intrinsically stretchable materials has emerged as a prominent strategy for achieving stretchability in transistors and functional circuits[16,20–26]. However, these materials often exhibit inferior electrical properties, thermal/chemical instability, and incompatibility with complementary-metal-oxide-semiconductor (CMOS) processes. While hybrid integration methods have been used to attach silicon-based electronic onto stretchable substrates, they still rely on costly manufacturing processes[27] or use bulky conventional bulky chips[28,29].

Recently, metal-oxide semiconductors have been considered as promising material candidates for emerging stretchable electronics as listed in Supplementary Table 1, owing to high carrier mobility, excellent stability, and well-established and matured large-scale processes[30–34]. The development of stretchable electronic fabrication techniques, including metal-oxide TFTs, provides a more sustainable and cost-effective long-term solution. However, their utilization in stretchable electronics has been hindered by the inherent rigidity and low fracture toughness. Numerous prominent studies have demonstrated that structural design and strain engineering methods such as wavy structures[30,32,34–36], mesa-shaped geometries[37,38], and rigid island structures[39–41] can enable high stretchability with good electrical performances. Especially, the rigid island approach has been considered as one of the successful strategies for strain isolated device integration, constructing transistors on rigid island surfaces and connecting them through stretchable interconnects[42,43]. With appropriate structure design, metal-oxide based electronic devices can be a promising platform for stretchable electronic systems with excellent electrical properties and mass productivity. Although the previous advances on the rigid island-based stretchable electronics are noteworthy, they often present a unit device or simple device array with low integration density owing to complex wiring electrode structures and lack of integration strategies on the intrinsically stretchable substances. In addition, the wavy or serpentine structures often used in conventional silicon-based stretchable inorganic devices[27,35,41] not only reduce device density, but also constrain the freedom to design stretchable electronic systems. This limitation hinders the design of complex and multifunctional systems in stretchable electronics. The realisation of practical stretchable electronics, including displays and sensor arrays, therefore faces challenges related to improving fabrication processes, yields and transistor densities. To overcome these limitations, particularly in terms of full integration of stretchable device and circuitry for viable industrial applications, the following issues must be addressed. The discrepancy of mechanical/chemical properties between the rigid and non-rigid (stretchable) regions on an elastic substrate can cause interconnection or device failure due to significant strain concentration at the areal interface. Secondly, for the successful application of this approach to the area-scalable and high-density electronics, it is crucial to introduce high-precision patterning of rigid islands, ideally utilizing CMOS compatible process. Additionally, reliable structures and materials for interconnection (wiring) electrodes are strongly required to achieve highly integrated and large-area scalable stretchable electronics while maintaining a planar structure for design flexibility[35,38,40,41].

Here, we demonstrate a versatile approach for full integration of highly stretchable metal-oxide transistors and circuitry with large-area scalability. The molecular-tailored elastomer substrates enable strong adhesion with the photo-patterned polymeric rigid island, successfully preventing delamination at the island/substrate interface and avoiding strain-induced device and interconnection failure. More importantly, full integration with high definition can be achieved by employing monolithically integrated liquid metal fluidic interconnects with dual-island metal-oxide transistors, forming embedding architecture in the elastic substances. Our approach not only maintains compatibility with conventional processes used on rigid substrates but also offers a high degree of freedom in circuit design. Furthermore, in-depth investigation and optimization of the dual-island structure were performed through mechanical simulation using finite element analysis (FEA) along with electrical analysis. Finally, to ensure the viability of such platform for realizing more practical stretchable electronics, we demonstrate highly stretchable (strain up to 50%) metal oxide-based 7 × 7 matrix transistor arrays, various logic gates, and 7-stage ring oscillator circuits. Our work goes beyond the implementation of simple stretchable IGZO TFTs; it presents an effective stretchable platform for various metal-oxide TFT based electronics with an in-plane structure that enables monolithic processing.

## Results

### Monolithic fabrication of stretchable inorganic transistors

The fundamental design rule of our stretchable electronics is the arrangement of strain-resistant active elements in an elastic substrate with a dual rigid island architecture. (Fig. 1a, b). In particular, photolithography-based bottom-up approach was employed in consideration of mass production and the full fabrication process is addressed in detail in Methods and Supplementary Fig. 1. One of the significant factors that hinders high stretchability in rigid island/elastomer system is poor adhesion between soft elastomer and hard rigid island segments. For example, when subjected to a critical strain, interconnection or device failure can occur as the substantial stress concentrates at the soft/hard interfaces, leading to physical delamination[40]. The central idea of our study is the deployment of acrylic-crosslinking chemistry, leveraging acrylate functional groups specifically derived from bisphenol A glycerolate diacrylate. These functional groups were judiciously incorporated into our chosen soft and rigid polymers polyurethane acrylate (PUA) and polyepoxy acrylate (PEA), respectively. While at a glance, PUA and PEA are polar opposites in terms of their mechanical properties—the former displaying a high degree of pliability and the latter, notable rigidity—we have exploited the unique characteristic of these polymers to form robust covalent bonds with one another. The potential for this bond formation is demonstrated in Fig. 1c. The PUA component of our material was synthesized from an isocyanate-terminated polyurethane precursor. The end product is a substance with an innate mechanical softness, a characteristic feature of urethane acrylate. Its double bond chemistry also provides a unique functionality, allowing for photo-curing[44]. Contrastingly, we utilized an alkaline-soluble epoxy acrylate for its dual property of photo-patternability and inherent rigidity[45]. The latter characteristic can be ascribed to the presence of a hard segment within its molecular structure. Interestingly, despite their stark differences, both PUA and PEA share a commonality in the form of identical methyl acrylate terminations, as depicted in Fig. 1c. The bonding process begins with photo-curing to create rigid islands. During this stage, the polymers' mobility gradually decreases as curing progresses, leaving behind unreacted acrylate groups[46]. As we move to the next step, photo-induced polymerization of the urethane acrylate layer, the remaining unsaturated groups in the epoxy acrylate are believed to form strong covalent bonds with the urethane acrylate. This crucial interaction, facilitated by the shared methyl acrylate terminations, elevates the cohesive force between the two materials, transcending the weaker interactions that typically arise from physical adsorption or hydrogen bonding.

Fourier transform infrared (FTIR) spectroscopy was employed to elucidate the chemical bonding characteristics between PEA and PUA. The quantitative analysis focused on the acrylate functional group's strength, with results presented in Supplementary Fig. 2. Initially, we examined the FTIR spectra for PEA (Supplementary Fig. 2a) and PUA

(Supplementary Fig. 2b) over various curing times, identifying an absorption peak at 1450 cm$^{-1}$. This peak is indicative of the acrylate group's double bonds among the observed spectral features (Supplementary Fig. 2c). For the quantification of peak intensities, baselines were meticulously established for each peak, and subsequent integral values were calculated from the fitted graphs. The coefficient of determination (R$^2$), indicative of the degree of concordance between the fitted graphs and the original FTIR data, consistently exceeded 0.99% across all samples (Supplementary Table. 2). This process can be validated to ensure the reliability of the integral values derived from the fitted graphs comparing to those obtained directly from the raw FTIR data. An observed trend was that the peak strength diminished with increased curing time for both polymers, suggesting a higher utilization of acrylate groups in the crosslinking process. Notably, the peak intensities for PEA and PUA converged at specific curing durations −5 min for PEA and 4 min for PUA under photocuring conditions − indicating an optimal curing time for both polymers. Further, FTIR results for PEA cured for 5 min, PUA cured for 4 min, and a combined PEA (1 min)/PUA (4 min) sample are presented in Supplementary Fig. 2c, d. These curing times were chosen based on the premise of complete polymerization for individual PEA and PUA samples, and for the PEA/PUA sample, it aligned with the photolithography process

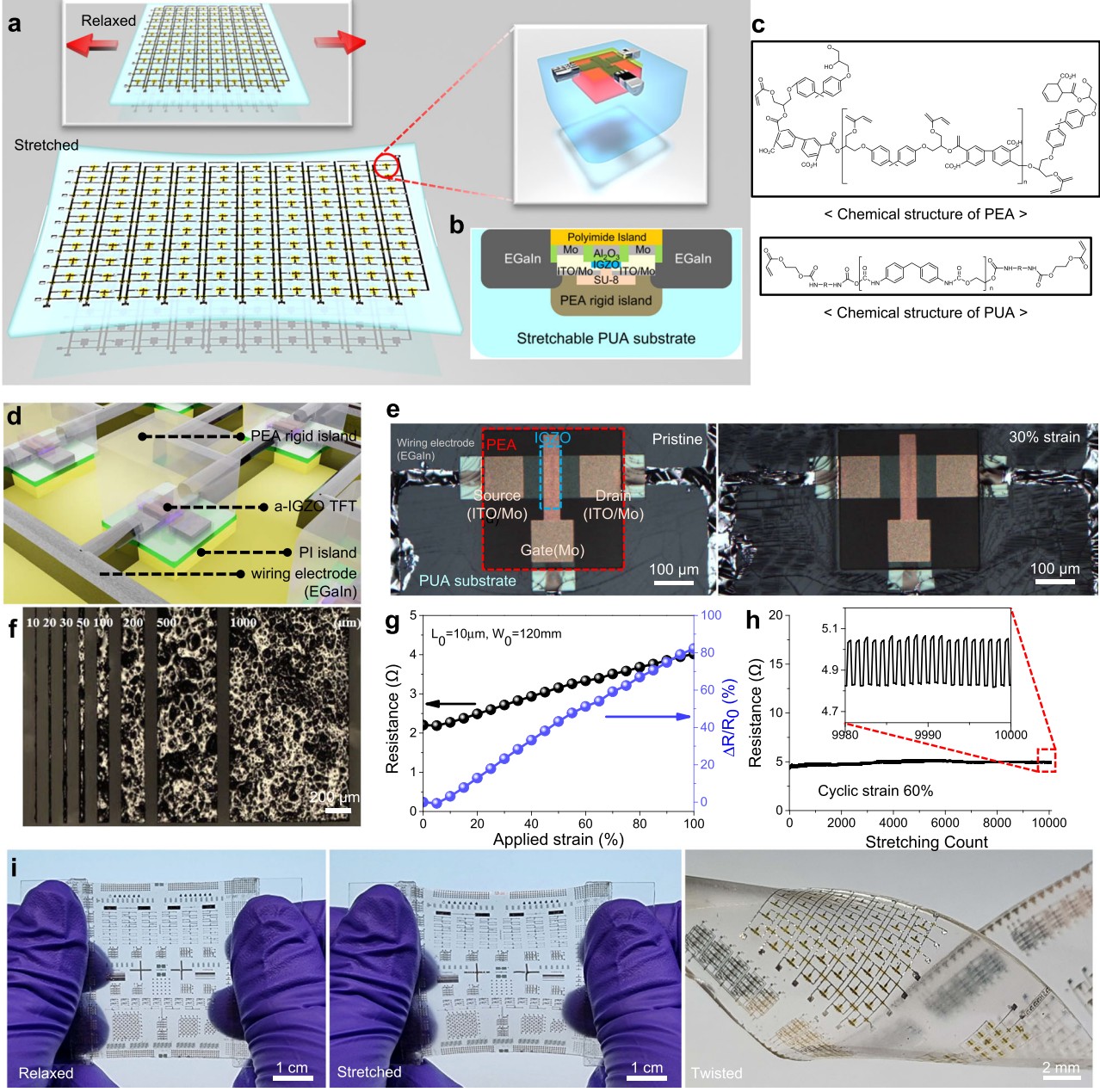

**Fig. 1 | Platform for stretchable metal oxide transistor array and circuitry. a** Schematics of stretchable metal oxide transistor array and device architecture. **b** Cross-sectional structure of stretchable transistor with dual rigid islands. **c** Chemical structures of urethane acrylate (UA) and epoxy acrylate (EA) oligomers for rigid island embedded stretchable substrate. An acrylic-crosslinking reaction initiated by ultraviolet light enables strong adhesion between the polyepoxy acrylate (PEA) and polyurethane acrylate (PUA) through strong covalent bonding. **d** Schematic illustration of amorphous indium-gallium-zinc-oxide ($a$-IGZO) transistors with eutectic gallium−indium (EGaIn) liquid metal interconnection. **e**, Optical micrographs of $a$-IGZO transistor under 0% (left) and 30% (right) strain. **f** Optical micrograph of patterned EGaIn liquid metal with different linewidths. **g** Electrical resistance variation of EGaIn liquid metal interconnects as a function of tensile strain. **h** Resistance variation of EGaIn liquid metal interconnects under 10,000 cyclic stretching (60% strain). **i** Photographs of stretchable $a$-IGZO transistor arrays and circuitry (~2000 transistors in ~5×5 cm²).

used in our study. For the PEA/PUA sample, PEA was partially cured (1 min) to retain acrylate double bonds, upon which PUA was coated and fully cured (4 min) to form a robust film, facilitating additional bonding with the residual acrylate in PEA. A marked difference was observed in the FTIR spectra of the combined PEA/PUA sample compared to the individual polymers. Integral area analysis of the acrylate group's double bond peak at 1450 cm$^{-1}$ revealed that the individual PEA and PUA had similar areas of 0.1561 and 0.1707, respectively, while the PEA/PUA composite showed a reduced area of 0.0863. This reduction suggests the formation of a strong covalent bond between the acrylate in PUA and the unsaturated groups remaining post-PEA photocuring, resulting in a robust stacked structure. This significant interaction, facilitated by methyl acrylate terminations, enhances the cohesive force between the materials, surpassing the weaker forces typically seen in physical adsorption or hydrogen bonding. To empirically evaluate the adhesive strength between PUA and PEA layers, we conducted a peel test on samples that integrated PUA with PEA in two distinct states: fully cured (for 5 min) and semi-cured (for 1 min). The test samples, prepared as per the design in Supplementary Fig. 2e using a silicone mold, demonstrated a marked difference in peel strength, affirming that photo-curing time control enhances the bonding force between the PEA and the stretchable PUA substrate (Supplementary Fig. 2f). The semi-cured PEA exhibited strong adhesion to PUA, preventing delamination until the PUA layer fractured. Conversely, fully cured PEA showed partial peeling at the interface, attributed to a lack of residual functional groups available for covalent bonding with urethane acrylate, as many acrylate groups were already consumed in the internal crosslinking of the polymer.

Furthermore, to achieve high-density integration, we implemented high-resolution patternable eutectic gallium–indium (EGaIn) liquid metal using CMOS-compatible lift-off process[47] as shown in Fig. 1d and Supplementary Fig. 3. In detail, to improve EGaIn wettability and prevent unintentional alloying of EGaIn with other metals such as aluminum and copper[48], molybdenum (Mo) was used as an adhesion layer that directly contacts to the EGaIn liquid electrode. Here, the moldable characteristics of EGaIn with a thin surface oxide layer allows high-resolution patterning with a minimum linewidth of ~10 μm via the photolithography process (Fig. 1f). The patterned EGaIn interconnection showed good stability under cyclic stretching tests owing to its unique combination of metal-level high conductivity and deformability as shown in Fig. 1g, h. Additionally, to prevent the stress penetration into the amorphous indium-gallium-zinc-oxide (a-IGZO) transistors by the thick liquid metal and protect the device upon stretching, polyimide (PI)/PEA dual island structure was utilized as shown in Fig. 1b, d. Using these processes, a high-performance stretchable a-IGZO transistor array with device density of 442 transistors/cm$^2$ (Supplementary Fig. 4) and excellent mechanical robustness (Fig. 1e, i) was fabricated.

## Material and structural designs for strain-resistant transistors and circuitry

As described, the adhesion between rigid island and stretchable substrate is important for achieving high stretchability. To verify the strong adhesion of molecular-tailored acrylate substrate, the critical strain for delamination was examined in various combinations of island/elastomer materials (Fig. 2a). The critical strain is defined as the strain where the delamination initiates. PI, PEA, and SU-8 were used to form square rigid islands of 350 μm × 350 μm, and PUA, PDMS, thermoplastic polyurethane (TPU), and styrene-ethylene-butylene-styrene (SEBS) elastomeric materials were used as the stretchable substrate. 10:1 and 20:1 PDMS are cases with different ratios to the curing agent, and as the ratio increases, the hardness of the material decreases. As shown in Supplementary Fig. 5, the rigid islands on the PDMS (10:1) substrate readily undergo delamination even with a slight strain of 10%, resulting in out-of-plane deformation perpendicular to the

applied strain. Moreover, when strains exceeding 20% are applied, additional openings at the edges become noticeable, further contributing to the out-of-plane deformation. When the strain was increased beyond 40%, the rigid islands became nearly fully detached from the substrate. In the case of PDMS (20:1), although there was a slight increase in critical strain due to the decrease in material rigidity with the change in ratio, delamination occurred at 40%. This suggests that the bonding force between the PDMS substrate and the rigid island is weak, making it highly susceptible to out-of-plane strain defects, where the rigid island easily peels off from the substrate. In contrast, the rigid islands on PUA substrate exhibited enhanced resistance to delamination in comparison to those on PDMS. PI and SU-8 islands on PUA began to be peeled off at around 40% and 60% strain, respectively. SEBS soft substrates also failed to maintain a strong bond with the rigid island, and delamination was observed at approximately 40%. In the case of TPU, it was observed that a TPU solution using dimethylformamide as a solvent caused significant damage to PEA when coated, and therefore, it is deemed unsuitable for the test. It is worth noting that PEA/PUA structure endured up to an 80% strain with no physical impairment such as fragmentation or delamination. This exceptional durability is primarily attributed to the potent adhesion between PEA and PUA, forged by the covalent bonds. Following the curing process, a considerable quantity of unreacted acrylate groups, or residual reactors, remain within the PEA islands. When the PUA layer is introduced, these residual reactors form covalent bonds with the acrylate groups in the PUA, resulting in a strong interfacial adhesion. This covalent bonding unites the PEA islands and the PUA matrix into a single entity, capable of withstanding high strains without structural failure.

Scanning electron microscopy (SEM) was employed to scrutinize the intersections between the islands and the elastomer, revealing details of the structural response when a strain is applied. Figure 2b, c display the SEM images of PEA/PUA substrate under 30% tensile strain. Notably, the images reveal an absence of fractures or delamination in the PEA/PUA substrate, underscoring its exceptional structural integrity under strain. In stark contrast, when examining the PEA/PDMS substrate under the same condition, a different scenario unfolds. This alternative substrate reveals the formation of voids, caused by the detachment of PDMS from the periphery of the PEA rigid islands. This detachment indicates a lack of bonding between the PEA and PDMS, as evidenced by the sharp separation surfaces observed in the micrographs. Upon further strain, at 90%, signs of delamination begin to appear even in the PEA/PUA substrate. However, the nature of this delamination is markedly different from that observed in the PDMS substrate. In Fig. 2c, where the PEA island is delaminated from PUA, an intriguing web-like structure is visible at the interface. This structure appears on both sides of the interface, suggestive of a gripped separation rather than a clean break. This observation leads to the hypothesis that the strong covalent bond between PEA and PUA does not allow for a simple, clean detachment under high strain. Instead, the interface is torn apart, leaving behind the web-like structure. This characteristic provides further evidence of the robust nature of the covalent bonds that link the PEA and PUA components. These results highlight the significance of the PEA/PUA structure in effectively mitigating the detrimental impact of rigid island delamination, particularly concerning the inter-island electrical connectivity, which plays a vital role in the overall functionality of stretchable electronic systems.

Further to find out the mechanical stress distribution under strain and the correlation between geometrical design and the mechanical stress, FEA simulations for PEA/PUA combination was performed. The thickness of PUA substrate was set to 200 μm and the effect of rigid island thickness was first examined as shown in Fig. 2d, e and Supplementary Fig. 6. In particular, the strain at the center of the rigid island and volume average stress of the island under 30% stretching

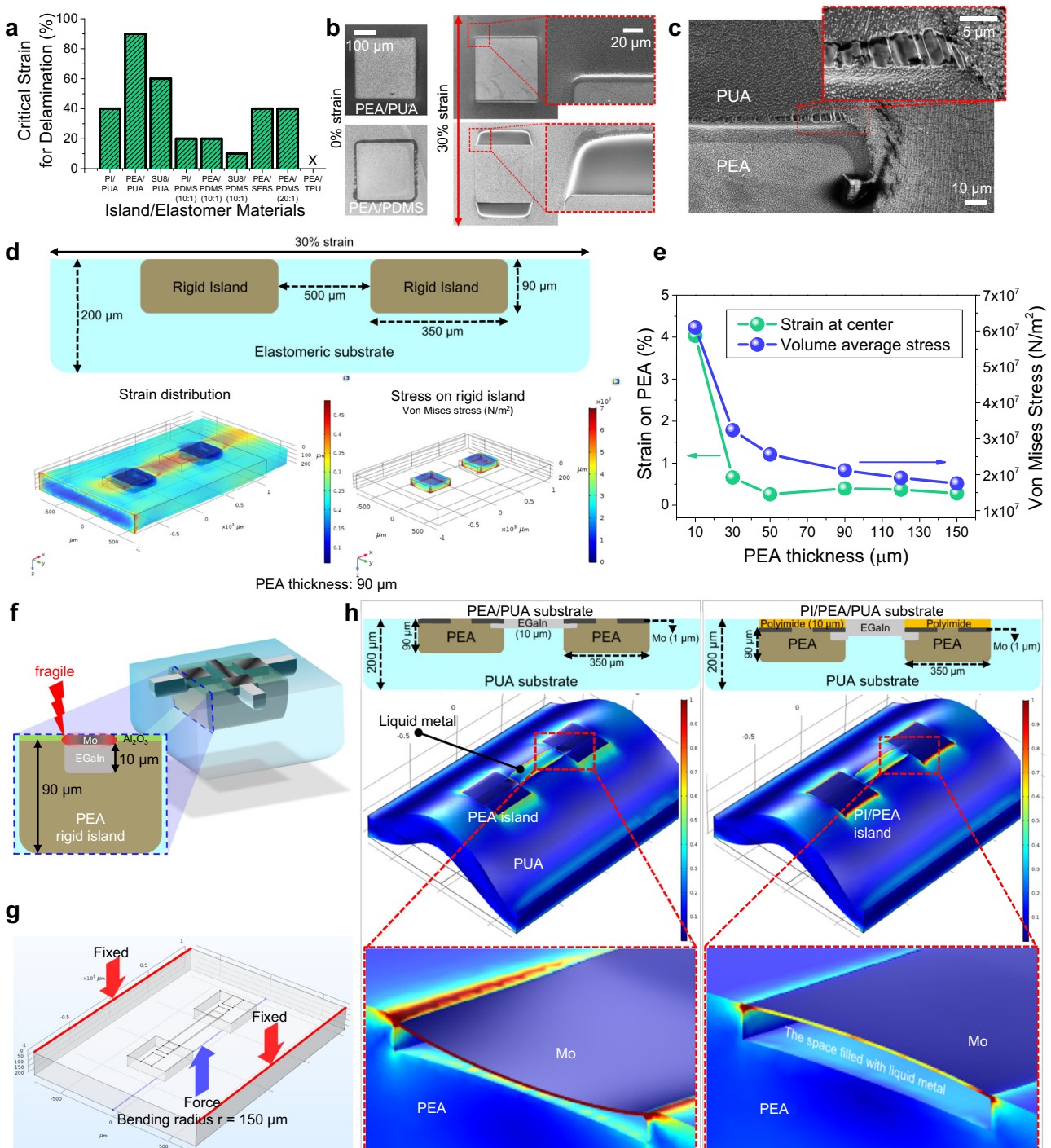

**Fig. 2 | Molecular-tailored elastomer substrate and geometry optimization.**
**a** Critical strains for delamination in different combinations of rigid island/elastomer materials. **b** Scanning electron microscopy (SEM) images of PEA/PUA and PEA/PDMS substrates under 0% and 30% strain. **c** SEM images of PEA/PUA substrate under 90% tensile strain. The PEA island is delaminated from PUA with a web-like structure at the interface. **d** FEA simulation for strain distribution and volume average stress of the PEA rigid island. **e** Simulated strain at the center of the rigid island and the volume average stress under 30% of tensile strain with different PEA thickness. **f** Schematic cross-section of PEA/PUA substrate with EGaIn interconnects. **g** Bending simulating condition for PEA/PUA substrate with EGaIn interconnects. **h** Strain distributions in PEA/PUA and PI/PEA/PUA substrates.

are analyzed (Fig. 2d). Since the center of the rigid island is where the *a*-IGZO transistor is located, strain at the center represents the approximate strain applied to the device. Also, the volume average stress is related to the mechanical stress that causes damage to the rigid island itself or delamination of island/elastomer interface. As shown in Fig. 2e, the volume average stress gradually decreases as the PEA thickness increases. Also, the strain at the center decreases rapidly as

the PEA thickness increases and nearly saturates after around 50 μm. In the case of strain at the center, there is a general trend of decrease with the increasing thickness of PEA across almost all regions. However, a slight increase is observed in the range from 50 μm to 90 μm, which is attributed to the deformation caused by the change in stiffness due to the thickness of the rigid island. As observed in the cross-sectional data in Supplementary Fig. 7, the rigid island deforms in a bending manner,

and the degree of bending is greater when the rigid island is thinner. Upon flexion, compressive strain is applied on one side, while tensile strain is applied on the opposite side. The area from which we extracted the strain at the center is subjected to compressive strain from bent shape, which can partially offset the tensile strain caused by the stretching applied to the global substrate.

Based on these results, the PEA rigid island was designed as thick as possible (90 μm) considering the capability of fine patterning. Next, the influence of spacing distance between neighboring two rigid islands was investigated (200 to 2000 μm). As shown in Supplementary Fig. 8, no substantial differences were observed in the stress distribution on the rigid islands. Considering that the devices are located only on the rigid islands, square shape and small spacing distance are then favorable for achieving a high device density. However, with a high density of islands, the areas to disperse the strain applied to the entire substrate are reduced, increasing the stress concentration at the rigid islands (Supplementary Fig. 9). Moreover, As demonstrated in Supplementary Fig. 10, even with the portion of rigid island region remained fixed, reducing the size of the rigid islands results in the reduction in the maximum stress on the components. Therefore, we covered the rigid island over individual device units by reducing its size, and this also greatly aids in enhancing the design flexibility of circuits and devices. The stress on the outer walls of islands are major causes of delamination and out-of-plane distortion under stretching which could be suppressed this phenomenon through quasi-homogeneous mechanical structure design based on acrylate-crosslinking chemistry (PEA/PUA). Furthermore, the embedded architecture of our stretchable transistors has been analyzed to exhibit more resilient mechanical characteristics when compared to other similar structures utilizing rigid islands as shown in Supplementary Fig. 11. Considering all these constraints, we designed the embedded rigid islands with a square shape and thickness of 90 μm, and the island-to-island spacing larger than 200 μm.

After implementing the PEA/PUA structure and design optimization, a stretchable platform for single transistor was constructed. Beforehand, further investigation was conducted on the stretchable EGaIn interconnection. As mentioned above, the EGaIn has a thickness of ~10 μm which is much thicker than those of inorganic layers constituting the transistors. Additionally, the EGaIn interconnection should pass through the sides of PEA island to electrically connect the transistors. Particularly, as shown in Fig. 2f, about 1.6% of the PEA volume is filled with EGaIn which is in liquid phase with diminutive elastic modulus[47,49]. Also, the source/drain (S/D) electrodes in this region, with thickness of ~120 nm, have no structural support. Furthermore, this region is close to the interface between rigid island and soft elastomer substrate where the strain is highly concentrated due to large difference of elastic moduli. Supplementary Fig. 12 shows the a-IGZO transistors fabricated on a PEA/PUA substrate. Without the EGaIn interconnection, no significant mechanical damage was observed even under 50% of tensile strain (Supplementary Fig. 12a). However, with the EGaIn formed on the rigid island, severe damage was observed as shown in Supplementary Fig. 12b, even without the stretching test. Very soft liquid metal with a thickness of a few microns directly contacts the thin-film device through the PEA, leading to significant stress penetration into the interior of the rigid island. Even before conducting the stretching test, the device is exposed to a certain degree of mechanical stress during the fabrication, particularly during the process of detaching the elastic substrate from the carrier glass. While it is significantly less than the applied tensile strain during actual measurements, various forms of mechanical deformation, including flexion and strain, do occur. Consequently, even minor deformations during the process can damage the stiff thin-film devices as shown in Supplementary Fig. 12b. From the mechanical simulation, it is noted that severe distortion occurs in the Mo region where the EGaIn is contacted (Fig. 2h–g). We investigated the strain distribution (with

bending radius of 150 μm) of the structure where Mo S/D electrodes are interconnected with a wiring liquid metal electrode as illustrated in Fig. 2g. When using PEA-only (single) island structure, the maximum strain applied in the Mo film was 1.47%, which indicates a large portion of external deformation is transmitted to the transistor, especially to the S/D metal via the soft EGaIn material. This can eventually lead to cracks in the Mo film causing a device failure. Interestingly, by providing a dual-island structure (PI/Mo/PEA) with an additional 10 μm-thick PI island onto the Mo metal, the strain applied to the Mo film was dramatically reduced to 0.83%. To understand more clearly, we extracted and visualized the structural deformation and strain distribution under such conditions as shown in Fig. 2h. The Mo film undergoes severe mechanical distortion and strain distribution in the single-island structure, whereas in the dual-island structure, the distortion and strain of the Mo film are effectively released, possibly due to dispersion of the induced stresses into the additional PI layer. Consequently, by introducing an additional PI layer (dual-island structures) on top of the transistor device, not only mitigates the strain in the unit device area but also significantly reduces the interfacial strain between the Mo S/D metal and the EGaIn wiring liquid metal electrodes, facilitating more robust device integration.

## Electrical characterization of stretchable transistors

To evaluate the mechanical stretchability of a-IGZO transistors fabricated on PUA substrate with PI/PEA dual-island structure, the transfer characteristics were analyzed. The mechanical stretchability was examined by measuring the transfer characteristics as a function of tensile strain, applied in both parallel and perpendicular to the channel direction (Fig. 3a). As shown in Fig. 3b, it is evident that the a-IGZO transistor withstands the strain without any apparent damage. As shown in Fig. 3c, the a-IGZO transistors exhibited carrier mobility, on/off ratio, threshold voltage ($V_{th}$) and subthreshold slope of 12.5 cm$^2$/Vs, $>10^7$, 1.1 V, and 117 mV/dec, respectively (at 0% strain), which are almost identical to those fabricated on a bare glass substrate (Supplementary Fig. 13). Further, the mechanical stretchability was examined by measuring the transfer characteristics as a function of tensile strain, applied in both parallel and perpendicular to the channel direction (Fig. 3a). As shown in Fig. 3b, it is evident that the a-IGZO transistor withstands the strain without any apparent damage. When stretched up to 50%, although a small variation in the transfer curves was observed (Fig. 3c, d), the carrier mobility, $V_{th}$, and subthreshold slope were stable as shown in Fig. 3e, f. The output characteristics of the a-IGZO transistor under tensile strain was also demonstrated in Supplementary Fig. 14. Furthermore, the mechanical stability under cyclic stretching was investigated with a tensile strain of 30% for 10,000 cycles. As shown in Fig. 3g, h, the a-IGZO transistors stably operated without significant failure even after 10,000 cycles of stretching and releasing for both directions. It is to note that as the stretching cycle is increased, a slight negative $V_{th}$ shift and increase in the on-current were observed. These can be attributed to the increase in donor concentration in a-IGZO channel caused by the mechanical stress that induces Fermi level shift towards the conduction band[37,50–52]. In particular, the mechanical deformation modifies the average spacing in the metal-oxide lattice, leading to a structural rearrangement which causes metal atoms in the amorphous metal-oxide films to lose oxygen atoms[53]. In metal-oxide semiconductors, the oxygen vacancies, whether singly ionized ($V_O^+$) or doubly ionized ($V_O^{++}$) are considered to be the primary sources of these excess charge carriers. Therefore, the increase in excess carriers results in a shift in the Fermi level. Comparing the transfer characteristics in Fig. 3g, h, it can be observed that the curves under tensile strain applied perpendicular to the channel direction exhibit larger variation than those with strain applied parallel. The channel layer of our devices has an aspect ratio of 100 μm/10 μm (channel width/length), giving it a longer vertical shape. In other words, the channels are situated more distantly from

the center of the rigid island along the vertical axis than along the horizontal axis. When the island undergoes mechanical stress, large stress is predominantly concentrated on the surface of the rigid island, perpendicular to the tensile direction, while the central region endures the least stress. Consequently, under vertical strain, a region of the channel undergoes comparatively higher stress. This results in a more pronounced change in the transistor characteristics with perpendicular strain rather than parallel strain.

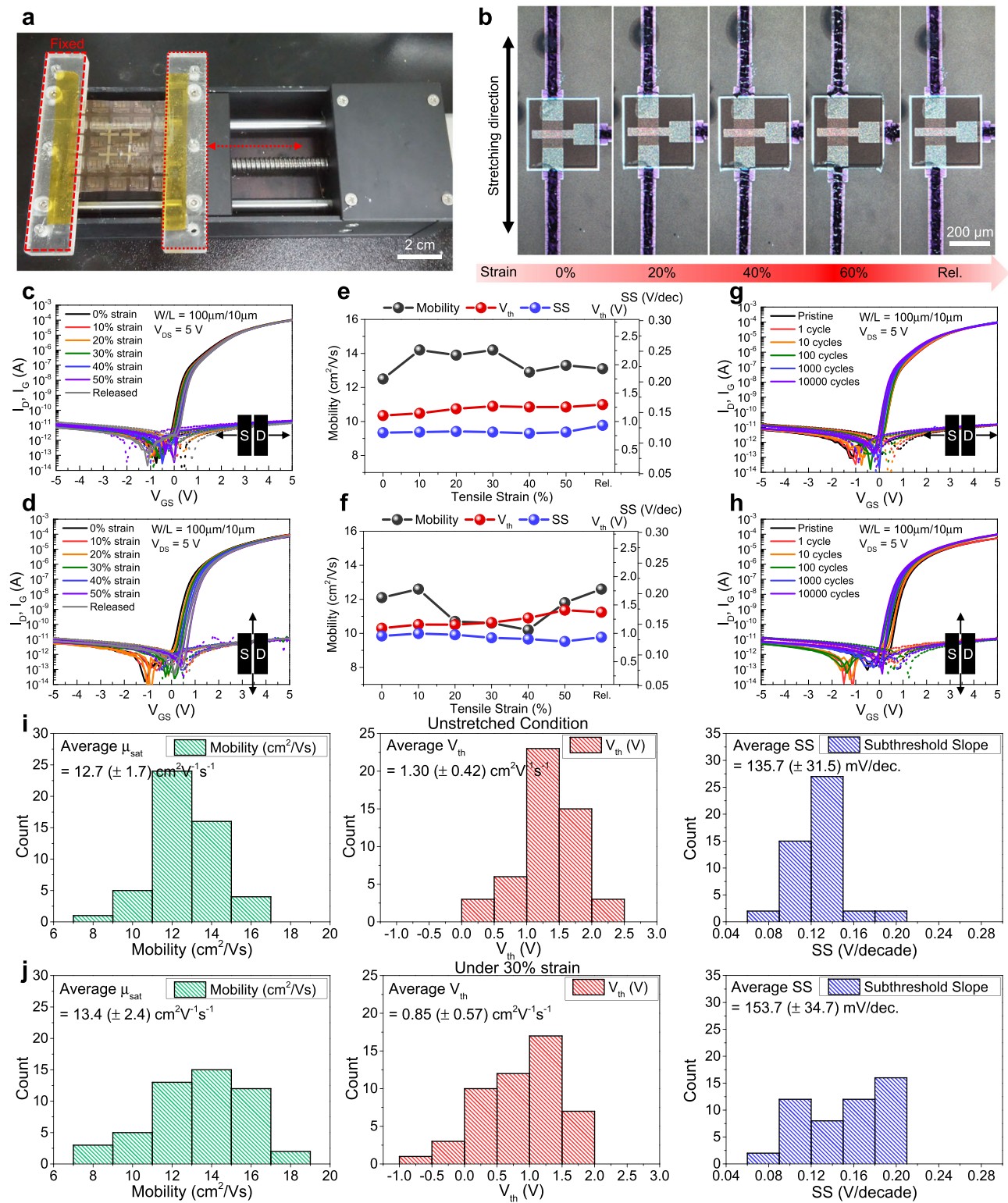

**Fig. 3 | Electrical characteristics of stretchable *a*-IGZO transistors under strain.** **a** Photograph of stretchable *a*-IGZO transistors on a stretching tester. **b**, Optical micrographs of *a*-IGZO transistor under stretching up to 60% and releasing. **c, d** Transfer characteristics of *a*-IGZO transistor as a function of tensile strain (**c**), parallel, and (**d**), perpendicular direction to the channel. **e, f** Variations of carrier mobility, threshold voltage, and subthreshold slope (SS) with tensile strain shown in (**c, d**), respectively. **g, h** Transfer characteristics of *a*-IGZO transistor during 10,000 stretching cycles (30% strain) (**g**), parallel, and (**h**), perpendicular direction to the channel. **i, j** Statistical data of carrier mobility, threshold voltage, and subthreshold slope in 50 stretchable *a*-IGZO transistors under (**i**), 0%, and (**j**), 30% strain.

To examine the uniformity for the large-area process, the statistical data of carrier mobility, $V_{th}$, and subthreshold slope were collected from 50 a-IGZO transistors for both under unstretched and stretched conditions (Fig. 3i, j). The unstretched transistors exhibited average saturation mobility of 12.7 ($\pm$1.7) cm$^2$/Vs with $V_{th}$ of 1.30 ($\pm$0.42) V and subthreshold slope of 135.7 ($\pm$31.5) mV/dec, presenting reasonably good uniformity. Under 30% stretching, although the standard deviation slightly increases, decent uniformity is maintained exhibiting average mobility, $V_{th}$, and subthreshold slope of 13.4 ($\pm$2.4) cm$^2$/Vs, 0.85 ($\pm$0.57) V, and 153.7 ($\pm$34.7) mV/dec, respectively. Additionally, various bias stress tests such as positive bias stress, negative bias stress, and positive bias illumination stress tests were conducted on the stretched a-IGZO transistors, and reasonably decent level of stability was observed as described in Supplementary Fig. 15. Moreover, as shown in Supplementary Fig. 16, we confirmed that our stretchable device maintains good bias stability under 30% of tensile stress.

To assess the durability of our stretchable devices under varying environmental conditions, we conducted various examination with our devices. First, to examine its temperature-dependent durability, we tested the patterned liquid metal and transistor characteristics under temperature variations, as shown in Supplementary Fig. 17. Both the liquid metal interconnection and the transistor exhibit decent properties with negligible differences at temperatures of 20 °C, 50 °C, and 80 °C. However, at −15 °C, a loss of stretchability was observed. This is due to the intrinsic property of EGaIn, which has a melting point of 15.5 °C, preventing it from maintaining its liquid state and causing the wiring electrodes to lose their stretchability and conductivity. Since some gallium-based liquid metals, such as Galinstan, have lower melting points (−19 °C), the resolution of this issue is feasible. To evaluate the durability of our device as a wearable, we tested its characteristics while it was attached to a finger of an artificial hand and immersed in liquid. We analyzed the transistor characteristics when the device was bent on the finger as shown in Supplementary Fig. 18a, and when immersed for 5 min in liquid, as illustrated in Supplementary Fig. 18b, c. Two types of liquid including deionized water and saline solution (0.9% NaCl solution) were used, with the saline solution employed to simulate the human sweat. Despite these diverse environmental changes, the device fully functioned without significant degradations (Supplementary Fig. 18d). Supplementary Fig. 18e shows the characteristics of the device stored in ambient conditions after the fabrication, confirming its proper operation even after 30 days. Thus, the excellent environmental endurance of our device demonstrates its potential for stretchable electronic platforms to the skin.

## High-density stretchable transistor array and integrated circuits

To illustrate the possibilities of utilizing highly stretchable a-IGZO transistors in large-area electronics, 7 × 7 a-IGZO transistor array, logic gates, and 7-stage ring oscillator were fabricated. As depicted in Fig. 4a, b, the transistors in our stretchable array share gate, drain, and source lines through liquid metal interconnections to simulate the active-matrix transistor backplane used in display devices. Within the transistor array comprising 49 transistors, the variation of transfer characteristics was small up to 50% strain. As demonstrated in the Supplementary Fig. 19a, no critical damage such as cracks, delamination, or wire breakage occurred in the array even at 50% elongation. This suggests that the strain is effectively reduced in the rigid regions and the stretchable EGaIn interconnects retain stable electrical conductivity over a large area during the mechanical deformation. Figure 4c, d show the stretchable inverter and ring oscillator consisting of 7 pseudo-NMOS inverters. The stretchable inverter was designed as a diode-load unipolar inverter based on two n-type transistors. Specifically, the inverter is implemented with a pseudo-NMOS configuration where the driver transistor (W/L = 100 μm/5 μm) and load transistor (W/L = 50 μm/20 μm) are connected in a series. As shown in Fig. 4c, the stretchable inverter exhibits typical logic input and output operations

in the voltage transfer curves (VTCs) even at strains up to 50%, although a slight negative $V_{th}$ shift is observed as the strain increases. We extracted the power consumption of the inverter using the expression, $I_{DD} \times V_{DD}$, and the power consumption of the inverter in the steady-state at $V_{in} = 0$ V and 5 V was 79.02 pW and 35.25 μW, respectively. Particularly high power consumption values were observed when a high input signal was applied, which is a drawback of pseudo-NMOS digital circuits with a diode-connected transistor serving as a pull-up transistor. At present, only NMOS transistors were used primarily due to the limitations of inorganic p-type transistors. Nevertheless, it would be possible to significantly enhance the power efficiency by utilizing to a CMOS structure, wherein high-quality p-type counterparts are incorporated. In the case of 7-stage ring oscillator, an oscillating frequency of 74.2 kHz was observed in the pristine state. At strain levels of 15% and 30%, the ring oscillator stably operated showing an oscillating frequency of 70.7 kHz and 67.7 kHz, respectively. Although a small decrease in oscillating frequency was observed with the strain, the result suggests that all the a-IGZO transistors comprising the ring oscillator fully function without failure under stretching. Although a small decrease in oscillating frequency was observed with the strain, the result suggests that all the a-IGZO transistors comprising the ring oscillator fully function without failure under stretching. When strain exceeding 30% was applied to the 7-stage ring oscillator, it was difficult to confirm the sound operation of the cascaded inverters. This malfunction appears to be due to the negative shift in the $V_{th}$ with increasing strain, as observed in the VTCs of each single inverter, rather than being caused by strain-induced damage. As indicated in Fig. 4c, the transition-triggering input voltage of the inverter is more negatively shifted with increasing strain. In the ring oscillator, the negatively shifted input low voltage across all the inverter stages of the ring oscillator causes the output voltage of the ring oscillator to gradually decrease by reducing the output voltage of every single inverter at 0 V. Therefore, as the input voltage of the inverter exceeds a critical threshold, a substantial reduction in the output range of each inverter produced, leading to a marked decrease in the amplitude of the output waveform or, in more severe cases, a complete inhibition of the oscillation. Those operational challenges in the ring oscillator may be more stringent on its inherent configuration with pseudo-NMOS structure, which can be overcome by the implementation of a CMOS structure via high-quality p-type pull-up transistors.

Additionally, stretchable logic gates including inverter, NAND, and NOR gates based on a-IGZO transistors are shown in Fig. 4e. Stretchable NAND and NOR gates were designed with a structure of two-driver and one-load transistors. For the NAND gates, two-driver transistors and one-load transistor were connected in a series. For the NOR gates, the two parallel driver transistors were connected serially to the load transistor. The logic gate functions were characterized by sweeping the input voltage from 0 to 10 V with $V_{DD}$ of 10 V. The input voltages of 0 and 10 V represent input logic states 0 and 1, respectively, for $V_{in,A}$ and $V_{in,B}$, respectively. The measured $V_{OUT}$ of the NAND gate shows logic state 0 only when both inputs ($V_{in,A}$ and $V_{in,B}$) are logic state 1. The measured $V_{OUT}$ of NOR gate shows logic state 1 only when both inputs ($V_{in,A}$ and $V_{in,B}$) are logic state 0. As shown in Fig. 4f and Supplementary Fig. 19b, all digital logic gates retained relatively accurate logical operation even under 50% strain. Particularly, in the case of the 50% strain condition, although the circuits showed no noticeable mechanical defects such as cracks or delamination, slightly deviated logical states of the circuits were observed. The decrease in the output voltage representing logic state 1 is caused by a negative $V_{th}$ shift that occurs in the driver transistors of the logic gate circuits due to the mechanical fatigue accumulated in the oxide semiconductors under significant stress as shown in Figs. 3g, h, and 4c. Consequently, these transistors do not fully turn off at an input voltage of 0 V, resulting in current leakage and a subsequent reduction in the output voltage when the logic state is 1. Nevertheless, our successful

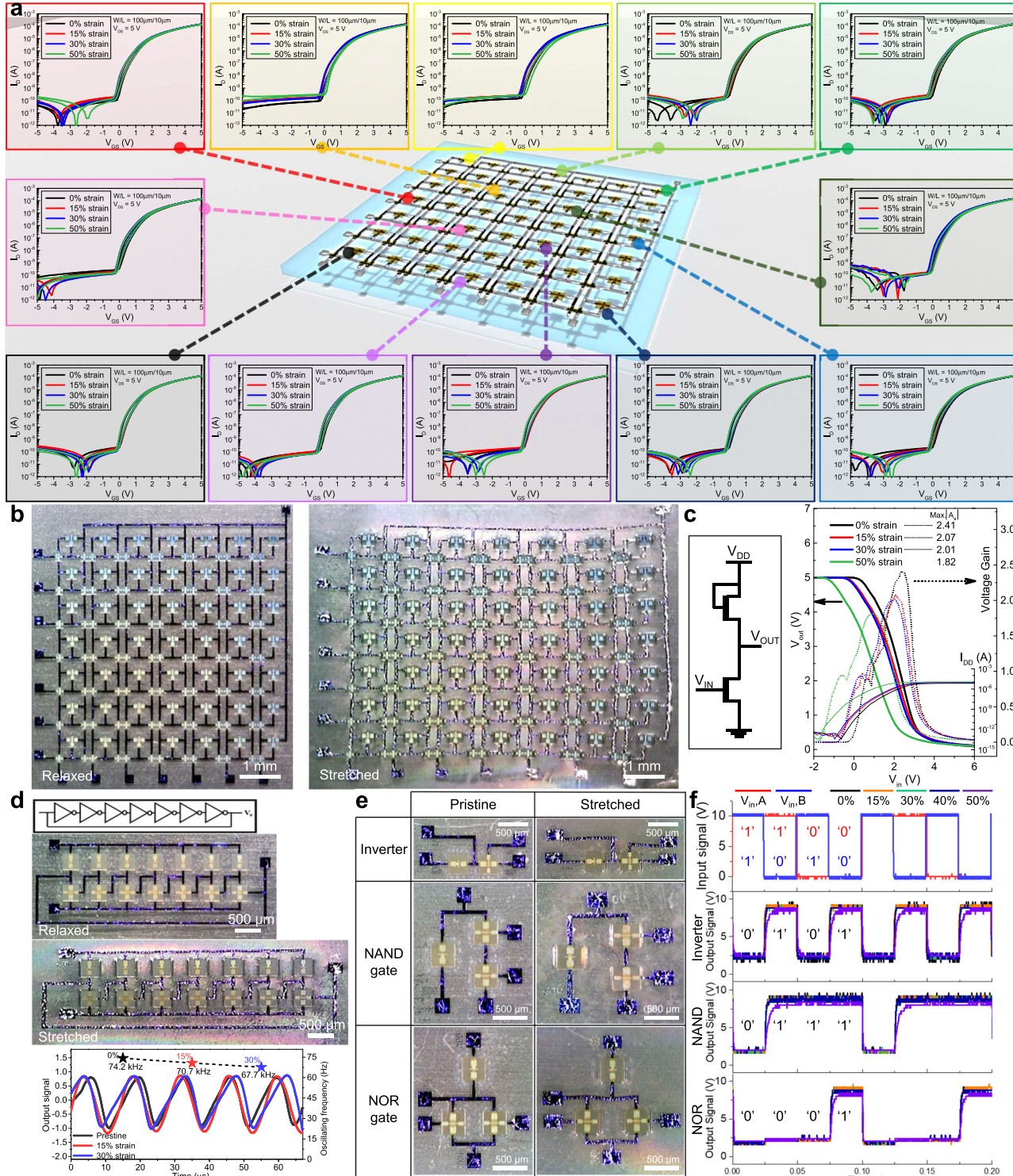

**Fig. 4 | Stretchable a-IGZO transistor array, logic gates, and 7-stage ring oscillator. a** Transfer characteristics of 7 × 7 stretchable *a*-IGZO transistor array (0%, 15%, 30%, and 50% strain). **b** Micrographs of 7 × 7 stretchable transistor array under 0%, and 30% strain. **c** Circuit diagram, output signals, drain current and voltage gain of stretchable pseudo-inverter. **d** Micrographs, output signals, and oscillating frequency of stretchable 7-stage ring oscillator under 0%, 15%, and 30% strain. **e** Micrographs of logic gates including inverter, NAND and NOR gates under 0% and 30% strain. **f** Input and output signals of inverter, NAND and NOR gates under 0%, 15%, 30%, 40% and 50% strain.

implementation of digital logic circuits on the stretchable platform demonstrates proper logic functionality. These results strongly corroborate that the stretchable *a*-IGZO transistor-based integrated circuits monolithically integrated with dual-island structure, EGaIn liquid metal interconnection, and the molecular-tailored PUA substrate could be extended to various stretchable electronic platforms with highly complex integrated digital circuits.

## Discussion

Here, we demonstrate the highly stretchable metal-oxide transistor arrays and circuitry using dual-island structure on molecular-tailored

elastomer substrates employing a CMOS-compatible fabrication process. The dual-island structure accompanied by monolithically integrated EGaIn interconnection on molecular-tailored elastomer substrates exhibited enhanced mechanical properties, allowing to sustain the electrical properties under a large strain. Furthermore, the successful demonstration of high-density transistor arrays, logic gates, and ring oscillator robustly validates the applicability in high-performance stretchable electronics. We envision that this versatile strategy of fabricating stretchable devices will advance the development of commercial-level stretchable electronics requiring high device density, logic functions, and large-area processibility. At the current research stage, challenges in reducing the size of rigid islands hinder the achievement of high-density integration of TFTs and integrated circuits, comparable to commercial-grade electronics. Notably, state-of-the-art oxide TFTs have reached sub-micrometer scales in applications such as augmented reality/virtual reality. The relatively large island size in our study is primarily due to TFT dimensions (100/10 μm channel width/length). Using conventional photolithography, we aim to reduce TFT dimensions and implement geometric designs for high-density integration, competing with commercial electronics. Another challenge can arise from the large thickness of stretchable EGaIn interconnects, hindering high-density integration. We envision that reducing the thickness through more elaborated coating processes will enable the integration of low-thickness EGaIn interconnects into high-density and mechanically robust stretchable electronics. Moreover, to address fully stretchable inorganic electronics, intrinsically stretchable semiconducting and insulating inorganic materials or device architecture should be explored, which may position more durable and densified fully stretchable, outperformed electronics in the applications. We envision that this versatile strategy of fabricating stretchable devices will advance the development of commercial-level stretchable electronics requiring high device density, logic functions, and large-area processibility.

## Methods

### Synthesis of EA and preparation of photoresist resin
A round-bottom flask was charged with bisphenol A glycerolate (BPA) diacrylate (7.65 g, 16.3 mmol) and propylene glycol monomethyl ether acetate (PGMEA) (15.8 g). The mixture was stirred until the solution became clear. 4,4-biphthalic anhydride (2.4 g, 8.15 mmol) and triphenylphosphine (cat.) were added to this mixture. This mixture was stirred at 100 °C for 4 h. Then, the mixture was supplemented with cis-1,2,3,6-tetrahydrophthalic anhydride (1.24 g, 8.15 mmol). The solution was stirred at 100 °C for a further 3 h. After cooling to room temperature, the solution was used without further purification. To prepare the negative-type photoresist (PR) resin, the PGMEA solution of EA was mixed with dipentaerythritol hexaacrylate (DPHA) and Irgacure 754 in the ratio of 10:2.1:0.38.

### Fabrication of flexible and stretchable metal-oxide electronics
Metal-oxide transistors and integrated circuits were manufactured on a flexible polyimide (PI) film as follows. First, a polyimide solution (Polyzen 150, PICOMAX) was spin-coated on a glass substrate and fully cured at 300 °C for the PI film (~3 μm) formation. One more layer of polyimide (PSPI 500PH, PICOMAX) is coated on PI glass, and the upper polyimide layer is photo-patterned in the shape of rigid island. To fabricate the gate electrode, Mo layer (50 nm) were deposited by RF-sputter. The gate electrodes were patterned by photolithography and wet etching. As the gate dielectric, $a$-Al$_2$O$_3$ layer (50 nm) was deposited by ALD at 150 °C using trimethyl aluminum as Al precursor and water as the O$_2$ source. The $a$-IGZO was deposited on the gate dielectric by RF-sputtering. The ITO (70 nm) and Mo (50 nm) S/D electrodes were sputtered and patterned by lift-off process. Su-8 photoresist (SU-8 3005, Microchem) was spin-coated and developed to passivate the $a$-IGZO semiconductor. Eutectic-gallium-indium (Sigma-Aldrich) stretchable electrode was coated by hand-roller on the lift-off layer made of negative photoresist. Lift-off was conducted by immersing in acetone for 2 h and softly spraying by acetone. After full fabrication of $a$-IGZO transistors and circuits on the polyimide substrate, the synthesized EA-based photoresist resin was spin-coated, followed by a typical photolithographic procedure involving UV exposure and development to produce cured EA (PEA) patterns. Then, to form the stretchable polymer (PUA), the solution of the synthesized UA compound was spin-coated and UV cured. The resulting structure was subsequently immersed into cold water (25 °C) to induce hygroscopic swelling of the polymers, allowing it to be peeled off safely from the glass. Finally, the polyimide layer was removed by oxygen plasma. (Supplementary Fig. 1).

### Finite element analysis
The mechanical simulation for substrate engineering and stress analysis were performed by finite element analysis (FEA) using COMSOL Multiphysics 5.3. The physical properties of the materials were employed in COMSOL Multiphysics, with the specific parameters provided in Supplementary Table 3 of the Supplementary Information.

### Device characterization
All the measurements were performed in air and dark ambient. Electrical characterization of the transistors and integrated circuit was performed by using Agilent 4156 C semiconductor parameter analyzer. The capacitance of dielectric was obtained through inductance, capacitance, and resistance (LCR) meter Agilent 4284 A. For the seven-stage ring-oscillator and logic circuits, the electrical characteristics were measured through digital storage oscilloscope (TDS2014C, Tektronix). The microstructures and morphologies of the films were investigated using optical microscopy (BX53M, Olympus) and digital microscope (UM12, MicroLinks). FE-SEM images were taken through SIGMA (Carl Zeiss) with accelerating voltage of 5 kV. Fourier Transform Infrared (FT-IR) spectroscopy (IRTracer-100, Shimadzu) was utilized for investigating the chemical bonds and functional groups of polymers. The peel strength of the comprised PEA/PUA was evaluated by Universal Materials Testing (LR10K-Plus, Lloyd Instruments Ltd.).

## Data availability
Source data are provided with the paper. Additional data related to this work are available from the corresponding authors upon request. Source data are provided with this paper.

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

## Acknowledgements

This research was partially supported by the National Research Foundation of Korea (NRF) grant funded by the Korea government (MSIP) (No. NRF-2019R1A2C2002447), by the Engineering Research Center of Excellence (ERC) Program supported by National Research Foundation (NRF), Korean Ministry of Science & ICT (MSIT) (Grant No. NRF-2017R1A5A1014708), and by the Nano & Material Technology Development Program through the National Research Foundation of Korea (NRF) funded by Ministry of Science and ICT(RS-2023-00281346).

## Author contributions

S.K.P., Y.-H.K. and J.-W.K. conceived the idea designed the materials and experiments. S.-H.K. and J.-W.J. fabricated and characterized stretchable inorganic transistors and circuits. J.M.L., S.B.S., S.B.C., D.B., M.-G.K. and J.-W.K. assisted with the materials design and device characterization. S.-H.K. and S.M. carried out the FEA simulation. S.-H.K., J.-W.J., Y.-H.K., J.-W.K. and S.K.P. wrote the manuscript incorporating input from all the authors. S.K.P., J.K., Y.-H.K. and J.-W.K. reviewed and revised the manuscript. The authors participated in discussions regarding the findings and collectively endorsed the manuscript.

## Competing interests

The authors declare no competing interests.
