## [Peer Review File · Nature Communications]

REVIEWER COMMENTS

Reviewer #1 (Remarks to the Author):

The authors conducted a substantial amount of additional experiments in response to the reviewer's comments, significantly improving the quality of the manuscript. While some doubts about the novelty of the research idea have not been completely resolved, the impressive aspect lies in the reliable implementation of a high-performance stretchable array using widely applicable oxide semiconductors. I believe the manuscript deserves publication in *Nature Communications*. However, there are some minor concerns that should be addressed before publication, as noted below.

1. In my opinion, the advancement of this paper seems to stem not from the introduction of a new idea but rather from the versatility to reliably implement continuous oxide TFTs through the appropriate combination of various ideas. In this regard, it would be beneficial to highlight the limitations of this research, such as the difficulty in reducing the size of rigid islands beyond a certain level or the challenge of increasing TFT integration density beyond a certain level. Providing possible solutions for these limitations in the outlook section would be appreciated. Additionally, it would be helpful to envision the extent to which stretchable integrated circuits can be produced using this fabrication method.

2. Although the authors explained that the crosslinking reaction proceeded through FTIR analysis, it is important to note that FTIR is not inherently a quantitative analysis method. To ensure accurate quantification analysis at the same peak, the baseline needs to be precisely established. The current presentation raises doubts about whether this aspect was adequately addressed.

Reviewer #2 (Remarks to the Author):

Kang et al. have carefully revised their paper, entitled "Full Integration of Highly Stretchable Inorganic Transistors and Circuits within Molecular-Tailored Elastic Substrates on a Large Scale," incorporating the comments from the reviewers. Additionally, the authors conducted comprehensive strain tests under various practical conditions, including assessments of thermal and chemical stability, as well as bending tests on an artificial hand. This reviewer recommends publication of the paper contingent on the implementation of specific revisions. To enhance the emphasis on the addressed points mentioned in the response letter and also mentioned by both Reviewer 1 and Reviewer 2 regarding the novelty of the study, it is suggested to integrate these points more prominently into the manuscript.

Additional technical comment: The study demonstrated the performance of diverse digital logic gates, including the inverter, NAND gate, and NOR gate, exclusively under a 30% strain. Could the authors

provide insights into the rationale behind this specific strain level? To enhance overall consistency, it is recommended that the authors expand their demonstration to encompass the results of all digital logic gates at strains surpassing 30%, aligning with the methodology applied to the IGZO transistor array.

Point-By-Point Response Letter (NCOMMS-23-59039-T)

In the following, we addressed all the comments raised by the reviewers on our manuscript (NCOMMS-23-59039-T) in a point-by-point fashion. In response to each raised point, we analyzed the existing data, conducted the literature survey, and explained the corresponding results followed by the changes we have made in the revised manuscript (refer to blue letters) and supplementary information.

=====
Responses to Reviewer 1's comments
=====

The authors conducted a substantial amount of additional experiments in response to the reviewer's comments, significantly improving the quality of the manuscript. While some doubts about the novelty of the research idea have not been completely resolved, the impressive aspect lies in the reliable implementation of a high-performance stretchable array using widely applicable oxide semiconductors. I believe the manuscript deserves publication in Nature Communications. However, there are some minor concerns that should be addressed before publication, as noted below.

[Comment #1]

In my opinion, the advancement of this paper seems to stem not from the introduction of a new idea but rather from the versatility to reliably implement continuous oxide TFTs through the appropriate combination of various ideas. In this regard, it would be beneficial to highlight the limitations of this research, such as the difficulty in reducing the size of rigid islands beyond a certain level or the challenge of increasing TFT integration density beyond a certain level. Providing possible solutions for these limitations in the outlook section would be appreciated. Additionally, it would be helpful to envision the extent to which stretchable integrated circuits can be produced using this fabrication method.

[Response #1]

We truly thank you the reviewer for the valuable comments and thoughtful analysis of our research paper. The reviewer's suggestions regarding the limitations of our research and potential solutions are valuable, and we would like to address them in more detail. Firstly, at the current stage of our research, we acknowledge the challenges associated with reducing the size of rigid islands beyond a certain level. This limitation impedes the realization of high-density integration of TFTs and integrated circuits, comparable to those found in commercial-grade electronics. Notably, the channel size of state-of-the-art oxide TFTs has decreased down to a few micrometers, even reaching sub-micrometer scales in high-resolution electronics such as augmented reality/virtual reality applications. It is important to highlight that the relatively large size of the island

structure used in our investigation ($\sim 350\ \mu\text{m}$) is primarily due to the dimensions of TFT applied in our research (channel width/length of $100/10\ \mu\text{m}$, respectively). As the entire fabrication process align with conventional photolithography process, we envision that reducing TFT dimensions and implementing an appropriate geometric design for TFT structures would enable high-density integration of TFTs and integrated circuits, positioning them competitively with commercial-grade electronics. One obstacle for achieving high-density integration lies in the large thickness of the stretchable EGaIn, which is relatively thicker than those used in the industry ($<1\ \mu\text{m}$). Given that the EGaIn exhibits high electrical conductivity, we propose that reducing the thickness through the adoption of more elaborated coating processes which enable low-thickness EGaIn interconnect would facilitate successful integration into high-density and mechanically robust stretchable electronics. More importantly, to address fully stretchable inorganic electronics, intrinsically stretchable semiconducting and insulating inorganic materials or device architecture should be explored, which may position more durable and competitive fully stretchable high density and outperformed electronics in the applications.

Following the reviewer's comments, we have revised the Introduction to highlight that the strength of our work lies in the effective combination of various ideas to achieve the reliable implementation of continuous oxide TFTs. In the outlook section, we will emphasize the importance of ongoing research to address this challenge and suggest potential avenues for improvement, which involve exploring novel materials, revisiting the fabrication processes, or incorporating advanced design methodologies and device architectures to enhance integration density and stretchability. Once again, we appreciate your thoughtful comments, and we believe that addressing these points will strengthen the overall contribution of our research.

➡ Revised manuscript (Page 3, line 12)

“While hybrid integration methods have been used to attach silicon-based electronic onto stretchable substrates, they still rely on costly manufacturing processes²⁷ or use conventional bulky chips^{28,29}.”

➡ Revised manuscript (Page 3, line 17)

“The development of stretchable electronic fabrication techniques, including metal-oxide TFTs, provides a more sustainable and cost-effective long-term solution.”

➡ Revised manuscript (Page 4, line 3)

“In addition, the wavy or serpentine structures often used in conventional silicon-based stretchable inorganic devices^{27,35,41} not only reduce device density, but also constrain the freedom to design stretchable electronic systems. This limitation hinders the design of complex and multifunctional systems in stretchable electronics. The realisation of practical stretchable electronics, including displays and sensor arrays, therefore faces challenges related to improving fabrication processes, yields and transistor densities.”

➡ Revised manuscript (Page 4, line 14)

“Additionally, reliable structures and materials for interconnection (wiring) electrodes are strongly required to achieve highly integrated and large-area scalable stretchable electronics **while maintaining a planar structure for design flexibility.**^{35,38,40,41}”

➡ Revised manuscript (Page 5, line 2)

“Our work goes beyond the implementation of simple stretchable IGZO TFTs; it presents an effective stretchable platform **for various metal-oxide TFT based electronics** with an in-plane structure that enables monolithic processing.”

➡ Revised manuscript (Page 24, line 20)

“**At the current research stage, challenges in reducing the size of rigid islands hinder the achievement of high-density integration of TFTs and integrated circuits, comparable to commercial-grade electronics. Notably, state-of-the-art oxide TFTs have reached sub-micrometer scales in applications such as augmented reality/virtual reality. The relatively large island size in our study is primarily due to TFT dimensions (100/10 μm channel width/length). Using conventional photolithography, we aim to reduce TFT dimensions and implement geometric designs for high-density integration, competing with commercial electronics. Another challenge can arise from the large thickness of stretchable EGaIn interconnects, hindering high-density integration. We envision that reducing the thickness through more elaborated coating processes will enable the integration of low-thickness EGaIn interconnects into high-density and mechanically robust stretchable electronics. Moreover, to address fully stretchable inorganic electronics, intrinsically stretchable semiconducting and insulating inorganic materials or device architecture should be explored, which may position more durable and densified fully stretchable, outperformed electronics in the applications. We envision that this versatile strategy of fabricating stretchable devices will advance the development of commercial-level stretchable electronics requiring high device density, logic functions, and large-area processibility.**”

[Comment #2]

Although the authors explained that the crosslinking reaction proceeded through FTIR analysis, it is important to note that FTIR is not inherently a quantitative analysis method. To ensure accurate quantification analysis at the same peak, the baseline needs to be precisely established. The current presentation raises doubts about whether this aspect was adequately addressed.

[Response #2]

Thank you for your insightful comment. As you mentioned, quantitative analysis with FTIR can be difficult due to its inherent limitations. Nevertheless, various studies have used FTIR to compare and analyze the amount of chemical bonds according to the peak intensity. This means that it can be used as a relevant number for comparative purposes, even though it does not provide accurate quantitative values. In this study,

we aimed to persuade the crosslinking reaction in/between PUA and PEA through an approach that quantifies this relevant figure of merit. For this purpose, data were collected by maintaining a uniform film thickness of 50 μm for all analyzed samples to ensure consistency firstly. In addition, graph fitting was performed by employing OriginPro software to evaluate the collected data. A comparison was quantitatively provided by presenting the integral value of the fitted curve and all R^2 values, representing the optimization between the experimental data and the fitted data exceeded 0.99%. These high R^2 values imply that the fitting is well conducted, which cannot be obtained if the baseline is not properly established. Therefore, we cautiously ensure that the analyzed data might be validated with full consideration of the concerns. Therefore, we have further increased the reliability of the results by including the R^2 value, recognizing the reader's potential skepticism about the meaning of crosslinking reaction based solely on FTIR data. This parameter, which represents the percentage of agreement between experimental and fitted data, complements the values essential for enhancing the reliability of the results.

➡ Revised manuscript (Page 8, line 14)

“For the quantification of peak intensities, baselines were meticulously established for each peak, and subsequent integral values were calculated from the fitted graphs. The coefficient of determination (R^2), indicative of the degree of concordance between the fitted graphs and the original FTIR data, consistently exceeded 0.99% across all samples (Table. S2). This process can be validated to ensure the reliability of the integral values derived from the fitted graphs comparing to those obtained directly from the raw FTIR data.”

➡ Revised supplementary information (Table. S2)

Sample	UV-curing durations	R^2 index	Related figure
PEA	30 s	0.9935	Supplementary Fig. 2a
	1 min	0.9906	
	3 min	0.9924	
	5, 10 min	0.9928	
PUA	30 s	0.9968	Supplementary Fig. 2b
	1 min	0.9953	
	3 min	0.9921	
	4, 10 min	0.9938	
PEA/PUA	PEA (1 min)/PUA (4 min)	0.9925	Supplementary Fig. 2d
PEA	4 min	0.9961	
PUA	5 min	0.9955	

Table R1 R^2 values that indicate the degree of agreement between the fitted graphs and the original FTIR spectra regarding **Supplementary Fig. 2**.

=====

Responses to Reviewer 2's comments

[Comment #1]

Kang et al. have carefully revised their paper, entitled "Full Integration of Highly Stretchable Inorganic Transistors and Circuits within Molecular-Tailored Elastic Substrates on a Large Scale," incorporating the comments from the reviewers. Additionally, the authors conducted comprehensive strain tests under various practical conditions, including assessments of thermal and chemical stability, as well as bending tests on an artificial hand. This reviewer recommends publication of the paper contingent on the implementation of specific revisions. To enhance the emphasis on the addressed points mentioned in the response letter and also mentioned by both Reviewer 1 and Reviewer 2 regarding the novelty of the study, it is suggested to integrate these points more prominently into the manuscript.

[Response #1]

We sincerely appreciate your time and effort in reviewing our manuscript and providing your insightful and constructive comments. We are greatly pleased to hear your positive assessment of the substantial additional experiments that have been conducted to improve the overall quality of our manuscript. As the reviewer commented, we have revised the manuscript more prominently to further emphasize the addressed points regarding the novelty of the study.

➡ Revised manuscript (Page 3, line 12)

“While hybrid integration methods have been used to attach silicon-based electronic onto stretchable substrates, they still rely on costly manufacturing processes²⁷ or use conventional bulky chips^{28,29}.”

➡ Revised manuscript (Page 3, line 17)

“The development of stretchable electronic fabrication techniques, including metal-oxide TFTs, provides a more sustainable and cost-effective long-term solution.”

➡ Revised manuscript (Page 4, line 3)

“In addition, the wavy or serpentine structures often used in conventional silicon-based stretchable inorganic devices^{27,35,41} not only reduce device density, but also constrain the freedom to design stretchable electronic systems. This limitation hinders the design of complex and multifunctional systems in stretchable electronics. The realisation of practical stretchable electronics, including displays and sensor arrays, therefore faces challenges related to improving fabrication processes, yields and transistor densities.”

➡ Revised manuscript (Page 4, line 14)

“Additionally, reliable structures and materials for interconnection (wiring) electrodes are strongly required to achieve highly integrated and large-area scalable stretchable electronics while maintaining a planar structure for design flexibility.^{35,38,40,41}”

“Our work goes beyond the implementation of simple stretchable IGZO TFTs; it presents an effective stretchable platform **for various metal-oxide TFT based electronics** with an in-plane structure that enables monolithic processing.”

[Comment #2]

The study demonstrated the performance of diverse digital logic gates, including the inverter, NAND gate, and NOR gate, exclusively under a 30% strain. Could the authors provide insights into the rationale behind this specific strain level? To enhance overall consistency, it is recommended that the authors expand their demonstration to encompass the results of all digital logic gates at strains surpassing 30%, aligning with the methodology applied to the IGZO transistor array.

[Response #2]

We thank the reviewer for the valuable comments. Besides the electrical properties of a unit device, circuits performance can be often determined contingent on the device reliability and uniformity. Particularly, in the highly strained applications, the difficulty of forming coherent strain-resistant interconnection between devices and uneven strain on the entire substrates also lead to the degradation of circuit performance, which may encompass the failure or abnormal operation of the circuits under the harsh mechanical environments. The aforementioned rationales distinctly demonstrate that the realization of highly stretchable logic circuits is more stringent comparing to the unit device and array operation.

For digital logic circuits, more strain-relevant circuit operation was verified with a wide array of additional experiments under 40 ~ 50% strain conditions as shown in **Fig. R2**. Under the 50% strain condition, although the circuits showed no noticeable mechanical defects such as cracks or delamination (**Fig. R3**), slightly deviated output voltage representing logical state 1 of the circuits were observed. These results seem to be attributed to the negative shift of the transfer characteristics in the driver transistors by mechanical fatigue as a response to the increased strain (**Fig. 3g and 3h**). Consequently, these transistors do not fully turn off at an input voltage is 0V, resulting in leakage currents and a subsequent drop in output voltage when the logic state is intended to be 1, which leads to the deteriorated logic operation of the circuits in the strained conditions.

On the other hands, in the case of the 7-stage ring oscillator, it was difficult to confirm the sound operation of the cascaded inverters at strains exceeding 30%. This malfunction appears to be possibly due to changes in the transition points of each single inverter, against to the damage caused by the strain itself. **Fig. R1** demonstrates that as strain increases, the inverter input voltage where the transition occurs, is more negatively shifted. In the ring oscillator, the negatively shifted input voltage results in a narrower operational voltage margin through cascading of the inverter stages, leading to a marked decrease in the amplitude of the output waveform or, in more severe cases, an entire inhibition of the oscillations. Those operational challenges

in the ring oscillator may be more stringent on its inherent configuration with pseudo-NMOS structure, which can be overcome by the implementation of a CMOS structure via high-quality p-type pull-up transistors.

➡ Revised manuscript (Page 22, line 16)

“As shown in **Fig. 4c**, the stretchable inverter exhibits typical logic input and output operations in the voltage transfer curves (VTCs) even at strains up to 50%, although a slight negative V_{th} shift is observed as the strain increases.”

➡ Revised manuscript (Page 23, line 2)

“Although a small decrease in oscillating frequency was observed with the strain, the result suggests that all the α -IGZO transistors comprising the ring oscillator fully function without failure under stretching. When strain exceeding 30% was applied to the 7-stage ring oscillator, it was difficult to confirm the sound operation of the cascaded inverters. This malfunction appears to be due to the negative shift in the V_{th} with increasing strain, as observed in the VTCs of each single inverter, rather than being caused by strain-induced damage. As indicated in **Fig. 4c**, the transition-triggering input voltage of the inverter is more negatively shifted with increasing strain. In the ring oscillator, the negatively shifted input low voltage across all the inverter stages of the ring oscillator causes the output voltage of the ring oscillator to gradually decrease by reducing the output voltage of every single inverter at 0 V. Therefore, as the input voltage of the inverter exceeds a critical threshold, a substantial reduction in the output range of each inverter produced, leading to a marked decrease in the amplitude of the output waveform or, in more severe cases, a complete inhibition of the oscillation. Those operational challenges in the ring oscillator may be more stringent on its inherent configuration with pseudo-NMOS structure, which can be overcome by the implementation of a CMOS structure via high-quality p-type pull-up transistors.”

➡ Revised manuscript (Page 23, line 25)

“As shown in **Fig. 4f** and **Supplementary Fig. 19b**, all digital logic gates retained relatively accurate logical operation even under 50% strain. Particularly, in the case of the 50% strain condition, although the circuits showed no noticeable mechanical defects such as cracks or delamination, slightly deviated logical states of the circuits were observed. The decrease in the output voltage representing logic state 1 is caused by a negative V_{th} shift that occurs in the driver transistors of the logic gate circuits due to the mechanical fatigue accumulated in the oxide semiconductors under significant stress as shown in **Fig. 3g, h, and 4c**. Consequently, these transistors do not fully turn off at an input voltage of 0 V, resulting in current leakage and a subsequent reduction in the output voltage when the logic state is 1. Nevertheless, our successful implementation of digital logic circuits on the stretchable platform demonstrates proper logic functionality. These results strongly corroborate that the stretchable α -IGZO transistor-based integrated circuits monolithically integrated with dual-island structure, EGaIn liquid metal interconnection, and the molecular-tailored PUA substrate could be extended to various stretchable electronic platforms with highly complex integrated digital circuits.”

➡ Revised figure (Fig. 4c)

Fig. R1 a. Circuit diagram, output signals, drain current and voltage gain of stretchable pseudo-inverter.

➡ Revised figure (Fig. 4f)

Fig. R2 Input and output signals of inverter, NAND and NOR gates under 0%, 15%, 30%, 40%, and 50% strain.

➡ Revised supplementary information (Supplementary Fig. 19b)

Fig. R3 Micrographs of logic gates under 50% strain.

REVIEWERS' COMMENTS

Reviewer #1 (Remarks to the Author):

Authors addressed all the issues properly. Now the manuscript is ready for publication.

Reviewer #2 (Remarks to the Author):

The authors have diligently attended to all the comments provided by the reviewers, and I commend their efforts to elucidate and elevate the quality of the manuscript. I recommend accepting the manuscript in its present form.

Point-By-Point Response Letter (NCOMMS-23-59039A)

Below, it includes responses to the comments made by the reviewers on our manuscript (NCOMMS-23-59039A).

=====
Responses to Reviewer 1's comments
=====

Authors addressed all the issues properly. Now the manuscript is ready for publication.

[Response]

The excellent comments and suggestions previously provided by the Reviewer have been greatly helpful in enhancing the quality of the manuscript. We express our gratitude.

=====
Responses to Reviewer 2's comments
=====

[Comment #1]

The authors have diligently attended to all the comments provided by the reviewers, and I commend their efforts to elucidate and elevate the quality of the manuscript. I recommend accepting the manuscript in its present form.

[Response #1]

We sincerely appreciate the time and effort the reviewer has dedicated to providing insightful and constructive comments throughout the review process. The feedback has been invaluable in improving our work.